# Distribution of Epstein–Barr Virus LMP1 Variants in Patients with Infectious Mononucleosis and Association with Selected Biochemical and Hematological Parameters

**DOI:** 10.3390/pathogens12070915

**Published:** 2023-07-06

**Authors:** Snjezana Zidovec-Lepej, Margarita Batovic, Marija Rozman, Kristian Bodulić, Laura Prtorić, Ante Šokota, Andrea Nikcevic, Petra Simicic, Goran Tešović

**Affiliations:** 1Department of Immunological and Molecular Diagnostics, University Hospital for Infectious Diseases, 10 000 Zagreb, Croatia; marijarozman114@gmail.com (M.R.); petrasimicic@gmail.com (P.S.); 2Department of Medical and Laboratory Genetics, Children’s Hospital Zagreb, 10 000 Zagreb, Croatia; margi.batovic@gmail.com; 3Research Department, University Hospital for Infectious Diseases, 10 000 Zagreb, Croatia; kbodulic@bfm.hr; 4Pediatric Infectious Diseases Department, University Hospital for Infectious Diseases, 10 000 Zagreb, Croatia; laurapp3@gmail.com (L.P.); sokota.ante@gmail.com (A.Š.); andrea.nikcevic237@gmail.com (A.N.); 5Department of Infectious Diseases, University of Zagreb School of Medicine, 10 000 Zagreb, Croatia; gtestovic@bfm.com

**Keywords:** Epstein–Barr virus, LMP1, genotyping, co-infections, infectious mononucleosis

## Abstract

The molecular diversity of Epstein–Barr virus (EBV) is exceptionally complex and based on the characterization of sequences coding for several viral genes. The aim of this study was to analyze the distribution of EBV types 1 and 2 and to characterize LMP1 variants in a cohort of 73 patients with infectious mononucleosis (IM), as well as to investigate a possible association between viral diversity and relevant clinical parameters. Population-based sequencing of *EBNA-2* gene showed the presence of EBV type 1 in all IM patients. Analysis of *LMP1* gene found a restricted repertoire of LMP1 variants with the predominance of wild-type B95-8, China1, Mediterranean and North Carolina variants with the presence of more than one LMP1 variant in 16.4% of patients. Co-infections with different LMP1 variants were associated with significantly higher levels of C-reactive protein and lower levels of maximal neutrophil counts and minimal platelet count. The results of this study have shown a narrow repertoire of *LMP1* variants and an exclusive presence of EBV type 1 in a cohort of IM from Croatia, suggesting a characteristic local molecular pattern of this virus. The clinical importance of distinct immunobiological features of IM patients with LMP1 variant co-infections needs to be investigated further.

## 1. Introduction

Epstein–Barr virus (EBV) or human gammaherpesvirus 4 is a prototype oncogenic virus that belongs to the family *Herpesviridae*, subfamily *Gammaherpesvirinae* and genus *Lymphocryptovirus* [1]. This ubiquitous virus that infects more than 90% of the human population establishes a lifelong persistence in both immunocompetent and immunocompromised hosts [2]. Its replication cycle includes lytic and latent phases that are precisely regulated by the transcriptional modulation of viral genes within a specific microenvironment of the host. EBV infection is associated with considerable morbidity and mortality on the global level, both in the context of malignant and non-malignant diseases [2].

Acute infection with EBV usually occurs during childhood when it often remains asymptomatic, with infection in young adults and adolescents often leading to infectious mononucleosis (IM) [3,4]. EBV is mainly transmitted by exposure to oral secretion and exhibits a relatively long incubation period of about 6 weeks. Clinical presentation of IM usually includes fever, pharyngitis and lymphadenopathy that can be accompanied by maculopapular rash, abdominal pain, jaundice, splenomegaly, hepatomegaly, malaise, fatigue, nausea, vomiting, palatal petechiae as well as periorbital and eyelid edema [3,4]. Although IM typically presents as a benign self-limiting disease, complications of acute primary EBV infection may include airway obstruction due to oropharyngeal inflammation, meningoencephalitis, hemolytic anemia, streptococcal pharyngitis, thrombocytopenia with an exceptionally rare (<1% of patients) occurrence of conjunctivitis, hemophagocytic syndrome, myocarditis, neurological disorders, neutropenia, pancreatitis, parotitis, pericarditis, pneumonitis, psychological disorders and splenic rupture [4].

Due to the association of EBV infection with an increased risk for the development of several malignant diseases, as well as in the context of recent evidence on the association between EBV seroconversion and the development of multiple sclerosis, the effective prevention of EBV infection is of particular importance [5,6,7]. However, despite extensive pre-clinical and clinical research, including the recent implementation of mRNA-based technologies in prophylactic vaccine development, licensed EBV vaccines as well as effective antiviral treatments for EBV infection are still not available [8].

EBV genome is a linear, double-stranded DNA molecule that consists of 175 kb pairs, including 80 possible coding regions and coding for 44 microRNAs [2]. The variability of EBV genome is estimated at 0.002%, with mutations in selected latency genes contributing to the molecular diversity of the virus. Based on sequences coding for latent viral proteins (EBNA-2, EBNA-3A and EBNA-3C), the virus can be classified into types 1 and 2 that are characterized by distinct biological features, particularly in the context of T-cell biology [9,10].

An additional level of EBV classification complexity includes a sequence variation of the *LMP1* gene. The protein coded by *LMP1* gene is an essential viral oncogene that mimics the biological effects of CD40, that is responsible for the constitutive activation of several signal transduction pathways (including NF-κB, PI3K/AKT and JAK/STAT), that enable the survival and proliferation of B-cells and their subsequent differentiation into lymphoblastoid B-cells within a specific microenvironment [11,12]. Based on the sequence variation in the C-terminal part of LMP1 protein relative to the wild-type strain B95-8, EBV is classified into seven main variants with distinct geographical distribution: Alaskan (AL), China1, China2, China3, Mediterranean with (Med+) or without (Med−) deletions and North Carolina (NC) with several rare variants as well (SEA1, SEA2 and CAO) [13,14].

The molecular diversity of EBV (types 1 and 2, LMP1 variants) has been investigated in a variety of malignant diseases that are etiologically associated with EBV infection (including endemic Burkitt’s lymphoma, nasopharyngeal carcinoma, Hodgkin’s lymphoma, non-Hodgkin’s lymphoma, peripheral T-cell lymphoma, B-cell lymphoma), immunocompromised patients (transplanted persons, HIV-1-infected persons), healthy young adults as well as in men who have sex with men. However, the data on the clinical significance of these findings are limited [15,16,17,18,19,20,21,22,23,24,25,26,27,28,29,30,31,32]. Furthermore, the literature data on EBV heterogeneity in IM and possible association with clinically relevant parameters are scarce [26,28,33,34].

The aim of this study was to analyze the distribution of EBV LMP1 variants in pediatric IM patients from Croatia and to investigate a possible association between EBV genetic diversity and selected clinical/laboratory parameters in IM, with a particular emphasis on co-infections with different LMP1 variants.

## 2. Materials and Methods

### 2.1. Patients and Study Design

This study included 73 patients (42 males, 57.5%) with a clinical diagnosis of IM and >1000 EBV DNA copies/mL of peripheral blood that have been enrolled into clinical care at the University Hospital for Infectious Diseases “Dr. Fran Mihaljevic” (UHID), Zagreb, Croatia between 2015 and 2021. EBV molecular analysis was performed by using DNA aliquots from the bank of biological samples (leftover samples stored after routine clinical diagnostics was completed) available at the Department of Immunological and Molecular Diagnostics, UHID, Zagreb, Croatia.

Data on selected demographic (age; gender), clinical (hospitalization vs. outpatient clinics care vs. no admission; days of symptoms at the time of clinical presentation; maximum body temperature; duration of fever; clinical symptoms and complications; disease outcome) and laboratory parameters (routine hematological and biochemical parameters; quantification of EBV DNA from the peripheral blood) from a subgroup of 33 patients were extracted from patient’s electronic databases at UHID.

This study was approved by the Ethics committee of UHID on 28 August 2019 (no. 01-1247-3-2019).

### 2.2. PCR and Population-Based Sequencing

Determination of EBV types 1 and 2 was performed by using nested PCR with primers designed to enable the amplification of 497 bp specific for EBV type 1 and 150 bp specific for EBV type 2 as described by Mendes et al., 2008 (Table 1) [35].

Amplification of the 3′end of the *LMP1* gene was performed using primers OF 8081: 5′-GCTAAGGCATTCCCAGTAAA-3′ and OR 8744: 5′-GATGAACACCACCACGATG-3′ as described by Li et al., 2009 (Table 1) [36]. PCR reaction was carried out in 40 cycles at 94 °C for 30 s, 60 °C for 30 s and 72 °C for 1 min by using FastStart™ High Fidelity PCR System (Roche Diagnostics, Mannheim, Germany).

PCR products from nested PCR were used for population-based sequencing using BigDye™ Terminator v3.1 Cycle Sequencing Kit, (Applied Biosystems, Foster City, CA, USA). Obtained amplicons were purified with sodium acetate and 96% ethanol in the first series of centrifugation and with 70% ethanol in the second series, with adapted BigDye™ Terminator v3.1 Cycle Sequencing Kit protocol. SnapGene (version 6.2.1., GSL Biotech LLC, Boston, MA, USA) was used for the alignment of forward and reverse primers in order to obtain the consensus sequence. MEGA X (version 10.0.5., Mega Limited, Auckland, New Zealand) was used for the alignment of consensus sequences with the reference sequence (EBV strain B95-8, GenBank accession number: V01555.2). According to the mutations and an algorithm designed by Edwards et al., 1999, *LMP1* sequences (GenBank/EMBL/DDBJ database, MK507915-MK507954, OR115520-OR115550 and OR124062-OR124063) were classified into six variants: China1, China2, Alaskan, North Carolina, Mediterranean with deletion and Mediterranean without deletion or co-infections [13].

### 2.3. Phylogenetic Analysis LMP1 Sequences

Forty sequences of the 3′end EBV *LMP1* gene were primarily edited and assembled into contigs in Vector NTI software and are all available in the GenBank/EMBL/DDBJ database with accession numbers MK507915-MK507954. Those 448 bp *LMP1* nucleotide sequences were used for the construction and graphical representation of the phylogenetic tree in MEGA X software (version 10.0.5). To compare sequence homology, *LMP1* sequences were aligned with the reference sequences representing the six main EBV strains obtained from the GenBank/EMBL/DDBJ database under the accession numbers: B95-8 prototype strain (V01555), Med + with 30 bp deletion (AY337721), Med− without 30 bp deletion (AY493810), China1 (KC207813), Alaskan (AY337725) and NC strain (AY337726). All sequences were aligned using ClustalW method and the phylogenetic tree was constructed using the maximum likelihood (ML) method and GTR + G evolutionary model with the gamma distribution [37,38]. Branch lengths were calculated in the units of number of nucleotide differences per site, while the statistical significance of phylogeny was estimated by the bootstrap analysis with 1000 replicates. Aligned C-terminus sequences of LMP1 gene were translated to amino acid (aa) sequences and compared with a reference wild-type sequence B95-8 in CLC Sequence Viewer (version 8.0.0., QIAGEN, Redwood city, CA, USA) for the identification of characteristic aa changes and classification of EBV LMP1 variants, as previously described [13,39].

### 2.4. Statistical Analysis

Data visualization and analysis were performed in R (version 4.2.1., R Core Team 2022, Vienna, Austria) and ggplot2 package (version 3.4.2.) [40]. Distribution normality was assessed graphically and with the Shapiro–Wilk test. Numerical variables were reported using medians and ranges. Comparison between numerical variables was performed using the Mann–Whitney U test in case of two-sample comparisons and the Kruskal–Wallis test in case of multiple-sample comparisons. The association between categorical variables was assessed using the chi-squared test. All statistical tests were two-sided with a significance level of 95%.

## 3. Results

### 3.1. Distribution and Characterization of EBV LMP1 Variants in Patients with IM

The majority of IM patients were infected with EBV wild-type B95-8 (*n* = 24 patients, 32.9%), China1 (*n* = 15 patients, 20.5%) and the Mediterranean variant (12 patients with Mediterranean without deletion, 1 patient with Mediterranean carrying 69 bp deletion, 1 patient with Mediterranean carrying 30 bp deletion; *n* = 14 patients, 19.2%). LMP1 variant North Carolina was detected in eight patients only (11.0%). Co-infections between LMP1 variants were detected in twelve patients (16.4%, B95-8/China1 in eight patients and B95-8/North Carolina in four patients). Population-based sequencing of the *EBNA-2* gene showed the presence of EBV type 1 in all patients.

Forty sequences of *LMP1* C-terminus (168,726–168,213nt) that were characterized in detail for characteristic deletions, tandem repeats and aa substitutions were shown to cluster into four out of seven LMP1 variants: China1 (37.5%), Mediterranean (27.5%), B95-8 (25%) and North Carolina (10%) clusters by phylogenetic analysis (Figure 1).

The analysis of characteristic *LMP1* deletions (Table 2) demonstrated that the majority of Croatian IM isolates did not carry any deletions (57.5%), while 17 (42.5%) Croatian IM isolates exhibited deletions, the most common being a 30 bp deletion (40%), which corresponds to a deletion of 10 aa (346–355aa) [41]. Depending on their strain type, all clustered LMP1 isolates exhibited the corresponding deletion (China1) or its absence (B95-8 and North Carolina). When considering the Mediterranean cluster, the non-deleted Mediterranean subtype (22.5%) was the dominant subtype, while only two IM isolates (Cro_1861 and Cro_1807) were classified as deleted Med+ subgroup (5%), with Cro_1807 isolate being the only one with a rare 69 bp deletion between aa 333 and 355 [25].

The C-terminal domain of LMP1 protein has been shown to contain varying numbers of characteristic 11-aa tandem repeats between aa positions 250 to 298 [41]. Considering that the B95-8 prototype sequence has four of the stated repeats disrupted by 5 aa (275–279 aa) between the second and the third repeat, LMP1 isolates from this study were classified into two groups according to the presence of the stated repeat cluster. These two groups were equally represented within Croatian IM isolates, but their distribution was significantly different in the analyzed LMP1 variants (*p* < 0.001). The number of the 11-aa repeats within Croatian isolates varied from 3 to 7. All B95-8 clustered isolates carried the stated repeat cluster, while the majority of China1 (86.7%) and Mediterranean (54.5%) isolates contained 5 to 7 11-aa repeats. Interestingly, 7-aa repeats were detected in four isolates (17.5%), with three of them belonging to the China1 cluster and the remaining isolate belonging to the Mediterranean cluster.

### 3.2. Detection of Amino Acid Substitutions in LMP1 Gene in Croatian IM Isolates

The final part of LMP1 variant analysis included the identification of characteristic C-terminal aa substitutions in comparison to the reference wild-type sequence B95-8. Besides the seven previously described and well-established aa changes [41], additional 41 aa changes were identified in the Croatian IM isolates of this study (Table 3). All IM isolates belonging to the prototype B95-8 cluster had the seven characteristic aa mutations: S at position 229; L at positions 306 and 308; D at position 312; Q at positions 322 and 334 and G at position 344. Furthermore, additional 18 aa substitutions at 16 different positions were detected. Only the change at position 328 (E→Q) was present in all 10 B95-8 isolates, followed by position 212 (G→S/H) present in 80% of isolates. Moreover, two unique aa substitutions were detected for the first time for this prototype variant, one in Cro_1813 isolate at position 267 (P→R) and the other one in Cro_1809 isolate at position 321 (P→T).

The Mediterranean strain had the highest number of detected aa substitutions, 31 (64.6%) in total. A characteristic aa change for the Mediterranean strain present at position 229 (S→T) was detected in all eleven Mediterranean clustered isolates, while the substitution at position 322 (Q→E) was found in nine (81.8%) isolates. Among the rest of the twenty-nine identified substitutions, only three were found in all isolates and reference sequences. These substitutions corresponded to positions 309 (S→N), 334 (Q→R) and 338 (L→S). Substitutions of D to G at codons 255, 266, 282 and 293 represent the mutation of the second aa in the analyzed 11-aa tandem repeat units and were detected solely in Mediterranean variants in varying numbers of repeat units. Out of 10 isolates with the stated substitution, only Cro_1861 isolate carried the substitution in all four repeats, while Cro_1807 isolate carried the substitution in four out of five repeat units. The other eight isolates carried the stated substitution in the final two repeat units of the repeat cluster. Furthermore, two new aa substitutions for the Mediterranean variant were also detected for the first time in this study, in Cro_1827 isolate at position 231 (A→S) and in Cro_1822 isolate at position 315 (G→E).

All 15 China1 clustered isolates had characteristic changes at position 334 (Q→R) and, with the exception of Cro_1808 isolate, S at position 229. Twelve additional aa mutations were detected, with two of them (positions 309 (S→N) and 322 (Q→N)) found in all China1 isolates. Additionally, the substitution at position 309 (S→N) was shared between all LMP1 isolates, excluding the B95-8 strain. Moreover, substitution at position 338 (L→S) was detected in 13 (86.7%) China1 isolates, while four changes (213 (H→Q), 329 (N→I), 356 (D→N) and 361 (T→R)) were only found in individual China1 isolates. Finally, the complete North Carolina cluster had two characteristic changes at positions 306 (L→Q) and 322 (Q→T). An additional nine aa mutations were detected, with five of them being specific for the North Carolina strain; 250 (D→N), 313 (S→P), 331 (G→Q), 338 (L→P) and 352 (H→N).

### 3.3. Association between EBV LMP1 Variants and Selected Laboratory Findings in IM

In the following part of the study, we analyzed the possible association between EBV LMP1 variants and the clinical parameters of patients with IM. The main characteristics of the analyzed patients are shown in Table 4.

No significant association between the occurrence of IM symptoms in patients infected with different EBV *LMP1* variants was observed (*p* > 0.05). We also did not find significant associations between EBV LMP1 variants and patients’ laboratory parameters recorded on admission, including leukocyte and lymphocyte counts, C-reactive protein (CRP), hemoglobin, platelet count, levels of liver enzymes (aspartate aminotransferase, AST, alanine aminotransferase, ALT, lactate dehydrogenase, LDH and bilirubin (*p* > 0.05). Similarly, we did not observe significant associations between EBV LMP1 variants and maximal leukocyte, lymphocyte, neutrophil, or unsegmented leukocyte counts, levels of AST, ALT, LDH and bilirubin, or minimal levels of leukocytes, hemoglobin and platelets (*p* > 0.05).

We also evaluated the levels of selected laboratory parameters in patients exhibiting co-infection with different EBV LMP1 variants (Figure 2). Patients with EBV LMP1 variant co-infections had significantly higher levels of CRP on admission (medians 13.2 and 7.6 mg/L, *p* = 0.047) and significantly lower levels of maximal absolute neutrophil counts (ANC) (medians 2076 and 3001/μL, *p* = 0.014) and minimal platelet count (medians 172 and 202/μL, *p* = 0.042).

## 4. Discussion

In this study, we present the results on the distribution of EBV types 1 and 2, as well as LMP1 variants in a cohort of patients with EBV-associated IM receiving clinical care at the tertiary clinical center in Croatia. To our knowledge, this is the largest study on EBV molecular diversity in pediatric IM patients available so far. An analysis of EBV LMP1 variant distribution showed a narrow repertoire of variants, with a predominance of wild-type B95-8, China1, Mediterranean without deletion and North Carolina variants. Mediterranean LMP1 viral variants carrying 30 bp or 69 bp deletions were exceptionally rare. EBV LMP1 variants Alaskan, China2 and China3 were not detected in our IM cohort. Of note, the presence of more than one LMP1 variant was associated with selected hematological and biochemical parameters in IM patients. All sequenced isolates were classified as EBV type 1.

The majority of the literature data published so far investigated the distribution of EBV types 1 and 2, as well as the LMP1 variant distribution in cohorts of patients with various malignant diseases. However, the data on EBV variant distribution in IM are limited, highlighting the importance of this study [15,16,17,18,19,20,21,22,23,24,25,26,27,28,29,30,31,32].

Important biological differences between EBV types 1 and 2, particularly in their cellular tropism and oncogenic potential, have been described in various research models that require careful interpretation. Despite in vitro evidence showing a higher oncogenic potential of EBV type 1 (strain B95-8) compared with type 2 (strain AG876) in transforming B-cells into LCLs (lymphoblastoid cell lines), in vivo humanized mice model experiments as well as ex vivo data on LCLs cultivated from patients with endemic Burkitt’s lymphoma confirmed the oncogenic properties of EBV type 2 [41,42]. Interestingly, EBV type 2 exhibits a particular cellular tropism for T-cells as well [43,44].

Crawford et al., 2006, conducted a prospective study among 510 EBV seronegative university students from Edinburgh that have been followed for three years. This study demonstrated that EBV type 1 was significantly overrepresented in IM patients compared with students that seroconverted but did not develop IM [17]. In addition to showing that EBV type 1 infection is more likely to result in IM compared to EBV type 2 infection, this study identified penetrative sexual intercourse as a risk factor for EBV seroconversion [17]. A study by the same research group involving a university cohort of 2006 students reported that sexual activity in young adults represents an increased risk for type 2 EBV infection [18]. A study by Banko et al., 2016, from Serbia also showed an overrepresentation of EBV genotype 1 in IM patients from Europe (EBV type 2 infection was detected in one of thirty-two IM patients) [34]. The predominance of type 1 infections in IM has also been shown in an earlier study involving 30 IM patients [33]. In our study, all IM patients were infected with EBV type 1, which is consistent with the findings from the abovementioned studies [17,18,33,34].

*LMP1* gene diversity in IM patients from Europe has been analyzed in three studies so far [26,28,33]. Sandvej et al., 1994, showed that seven single-base mutations and the 30 bp deletion between codons of aa 322 and 366 in the *BNLF-1* gene were present not only in patients with malignant diseases from Malaysia and Denmark (Hodgkin’s disease, peripheral T-cell lymphoma), but also in IM patients from Denmark [26]. Furthermore, in a study analyzing *LMP1* gene diversity patients with a variety of malignant diseases, as well as in IM. Khanim et al., 1996, the presence of *LMP1* 30 bp deletion variant in IM patients of European descent was recorded (*n* = 8) [28].

Banko et al., 2012, analyzed LMP1 variant distribution in 30 IM patients with detectable plasma EBV DNA from Serbia (19/30 patients hospitalized, no demographic data available) and showed the highest frequency of wild-type and China1 variant (ten and nine patients, respectively) with North Carolina detected in six and Mediterranean variant in four patients [33]. Signature 30 pb deletion in the *LMP1* gene was detected in ten patients and a rare 27 bp deletion in one IM patient [33]. The results of the stated study clearly showed a geographic-associated pattern of LMP1 variants in IM patients from that particular region of Europe. EBV wild-type strain B95-8 is characterized by 4 repeats of 11 aa (positions 250–308) with a disruption of 5 aa within the third repeat. The majority of LMP1 variants in the IM patients described in this study were characterized by the stated repeat clusters (21 of 30 patients) [33].

The results of our study involving 73 patients from Croatia showed a narrow repertoire of LMP1 variants with the predominance of wild-type B95-8, China1, Mediterranean without deletion and North Carolina variants in IM patients. Importantly, this study demonstrated the presence of more than one LMP1 variant in a significant number of IM patients. Narrow patterns of EBV types and LMP1 variants observed in IM patients most likely, at least to a certain extent, reflect the homogeneity of the local population in that region of southeastern Europe. Therefore, the collection of additional data on EBV molecular diversity from IM cohorts from other countries and from patients of a different ethnic background is needed to provide a more complete landscape of EBV variation in this disease.

The only study showing an association between EBV molecular diversity and clinically relevant biochemical parameters in IM published so far involved 33 pediatric and adult IM. This study demonstrated a correlation between the characteristic combinations of *EBNA2*, *LMP1* and *EBNA1* polymorphisms (including deleted *LMP1*/P-thr and non-deleted *LMP1*/P-ala, genotype 1/4.5 33 bp *LMP1* repeats or genotype 2/4.5 33 bp *LMP1* repeats) and elevated levels of liver enzymes [33]. Of note, hepatic lesion is a relatively common clinical manifestation in IM that resolves spontaneously within several weeks in the majority of patients.

We also evaluated the impact of EBV molecular diversity on selected clinical and laboratory parameters of IM patients and have shown, for the first time, important differences in immunological parameters in IM patients with respect to the co-infection with multiple EBV variants. Notably, we did not find a significant association between EBV LMP1 variants and IM clinical presentation. This could result from other viral factors impacting the clinical presentation of IM, such as lytic gene expression and viral load.

Our results have shown significantly increased concentrations of CRP on admission in patients with LMP1 variant co-infections in comparison with single-variant-harboring IM patients. CRP is an acute-phase protein that is mainly synthesized by liver hepatocytes in response to the transcriptional activation of the *CRP* gene by inflammatory cytokines, including interleukin-6 [45]. Concentrations of CRP are increased in response to infection, injury and inflammation and mediated by regulation of the C1q component of the complement leading to pathogen opsonization that subsequently stimulates the synthesis of pro-inflammatory cytokines [46].

Elevated levels of CRP during the acute phase of EBV infection, e.g., IM, were found to be an independent risk factor for the development of fatigue over time by Pedersen et al., 2019 [46]. These findings have been substantiated by a study by Kristiansen et al., 2019, on 200 adolescents (12–20 years old) assessed 6 months after IM, showing significantly higher concentrations of serum CRP in the fatigued group of patients in comparison with those not experiencing fatigue [47]. Based on our results, we speculate that the presence of various immunogenic epitopes from multiple LMP1 variants might be associated with increased concentrations of pro-inflammatory cytokines that are involved in the transcriptional activation of the *CRP* gene, potentially leading to an increased CRP concentration.

Furthermore, we described significantly lower thrombocyte counts in IM patients with LMP1 variant co-infections. Transient thrombocytopenia is commonly described in patients with EBV-associated IM. A recent clinical review of 400 records of hospitalized patients (52.0% male; median age, 19 years (range, 15–87 years) by Páez-Guillán et al., 2023, showed that thrombocytopenia (platelet count ≤ 150 × 10^9^/L) was present in 29.7% of patients with severe findings (platelets <50 × 10^9^/L) present in 1.5% of patients [48]. Patients with thrombocytopenia showed distinct clinical and biological features with a lower frequency of typical mononucleosis symptoms such as sore throat and lymphadenopathy, a lower frequency of positive heterophil antibodies, a higher serum bilirubin concentration and prothrombin time, lower blood leukocyte and lymphocyte counts, lower concentrations of serum immunoglobulin G and immunoglobulin A and a larger spleen size [48]. The results of our study suggest that LMP1 variant co-infections might represent a novel biological feature of IM patients with thrombocytopenia.

In addition, significantly lower levels of ANC have been described in co-infected patients. However, mild neutropenia in IM is thought to be associated with transient bone marrow suppression often described in self-limited viral infections, and clinically severe cases are rarely described [49].

Studies on the molecular diversity of EBV in IM are of particular importance in the more detailed characterization of innate and specific immune responses to EBV, the fundamental immunobiology of EBV infection as well as in the development of prophylactic and therapeutic vaccines [50,51,52]. The recently described role of EBV as a co-factor in the development of multiple sclerosis has renewed the interest in the development of prophylactic based on mRNA technology [51]. In addition, the recently described candidate lipid-based LMP2-mRNA vaccine provides a scientific background for the development of novel cancer immunotherapy candidate vaccines [52].

In conclusion, our results have shown a narrow repertoire of *LMP1* variants and an exclusive presence of EBV type 1 in a cohort of IM from Croatia, possibly representing a characteristic regional molecular pattern. Distinct immunobiological features of patients exhibiting *LMP1* variant co-infection suggest that EBV molecular diversity might be of clinical relevance in IM. However, further studies on larger IM cohorts are required to resolve the issue of a possible association between viral strain specificity and IM severity.

## Figures and Tables

**Figure 1 pathogens-12-00915-f001:**
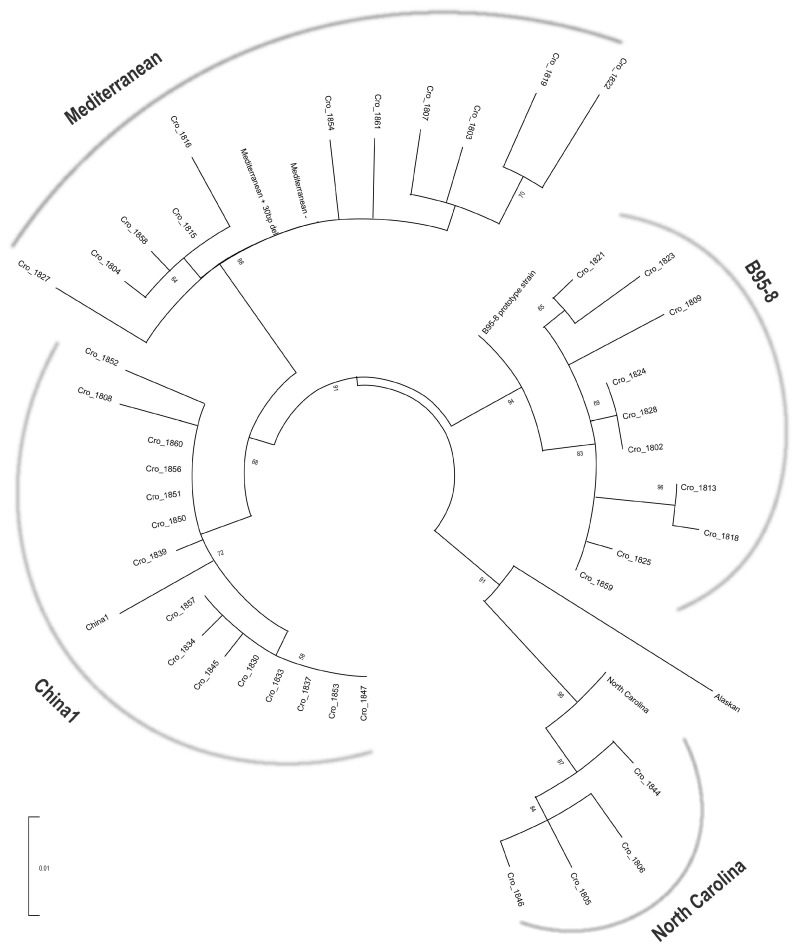
Phylogenetic tree of the C-terminus of *LMP1* nucleotide sequences. Phylogenetic analysis was performed using the ML method and GTR + G evolutionary model with the gamma distribution. Tree branch lengths were calculated in the units of number of nucleotide differences per site. The percentage of replicate trees in which associated taxa clustered (the bootstrap analysis with 1000 replicates) is shown next to the branches. Forty LMP1 sequences of Croatian IM isolates were segregated into 4 of 7 known LMP1 variant clusters: China1, Mediterranean, B95-8 and North Carolina.

**Figure 2 pathogens-12-00915-f002:**
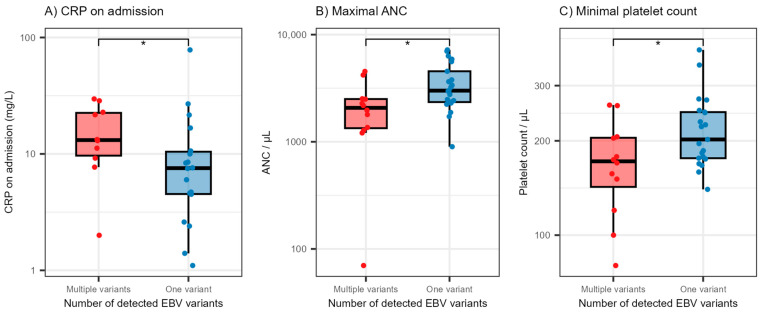
Distribution of C-reactive protein levels on admission (**A**), maximal absolute neutrophil count during hospitalization (**B**) and minimal platelet count during hospitalization (**C**) according to the number of detected EBV variants. The boxes show the median and interquartile range of the distribution, while the whiskers extend to the minimum and maximum non-outlier values of the distribution. Points denote individual participants. The y-axis is logarithmically scaled. * *p* < 0.05 (Mann–Whitney U test). EBV = Epstein–Barr virus, CRP = C-reactive protein, ANC = absolute neutrophil count.

**Table 1 pathogens-12-00915-t001:** Primers used for the amplification of *EBNA-2* and *LMP1* genes (as described by Mendes et al., 2008, and Li et al., 2009) [35,36].

Gene	Primers
*EBNA-2* (PCR)	F: 5′-AGGGATGCCTGGACACAAGA-3′ R: 5′-TGGTGCTGCTGGTG GTGGCAAT-3′
*EBNA-2* (nested PCR, EBV type 1)	F: 5′-TCTTGATAGGGATCCGCTAGGATA-3′ R: 5′-ACCGTGGTTCTGGACTATCTGGATC-3′
*EBNA-2* (nested PCR, EBV type 2)	F: 5’-CATGGTAGCCTTAGGACATA-3’ R: 5’-AGACTTAGTTGATGCCCTAG-3’
*LMP1*	F: 5’-GCT AAG GCA TTC CCA GTA AA-3’ R:5’-GAT GAA CAC CAC CAC GAT G-3’

EBV = Epstein–Barr virus, LMP1 = latent membrane protein 1.

**Table 2 pathogens-12-00915-t002:** Characteristic EBV *LMP1* deletions and tandem repeats in patients with infectious mononucleosis.

EBV *LMP1* Variant	Number of 33 Base Pair Tandem Repeats		Deletions in the *LMP1* Gene	
	3–4.5	5–7		
B95-8 Wild-Type	10	-	No deletion	10
China1	2	13	30 base pair deletions	15
Mediterranean	5	6	No deletion	9
			30 base pair deletion	1
			69 base pair deletion	1
North Carolina	3	1	No deletion	4
Total	20	20		

EBV = Epstein–Barr virus, LMP1 = latent membrane protein 1.

**Table 3 pathogens-12-00915-t003:** Amino acid substitutions found in LMP1 variants of Croatian patients with infectious mononucleosis. Four newly discovered aa mutations unique for Croatian IM isolates are marked.

aa Position	212	212	213	214	214	214	218	221	224	229	231 ^2^	240	248	250	250
B95-8 Prototype ^1^	G	G	H	E	E	E	N	E	H	S	A	Q	G	D	D
aa Change	S	H	Q	K	Q	G	T	D	Q	T	S	R	A	N	E
Number of Cases in Croatian IM Isolates															
B95-8	7	1	1	1									1		
China1	8		1	2	1			1		1					
Mediterranean	7					1	1	1	1	11	1	1			1
North Carolina	3													4	
aa Position	252	255	262	266	267 ^2^	270	282	293	306	309	313	314	315 ^2^	317	317
B95-8 Prototype ^1^	G	D	N	D	P	T	D	D	L	S	S	A	G	D	D
aa change	A	G	S	G	R	A	G	G	Q	N	P	G	E	E	N
Number of Cases in Croatian IM Isolates															
B95-8	1		1		1										
China1										15					
Mediterranean	4	1		2			5	9		11		1	1	1	1
North Carolina						1			4	4	4				
aa Position	319	321 ^2^	322	322	322	322	325	328	329	330	331	333	333	334	335
B95-8 Prototype ^1^	G	P	Q	Q	Q	Q	E	E	N	K	G	D	D	Q	G
aa Change	D	T	E	T	N	D	K	Q	I	T	Q	A	G	R	D
Number of Cases in Croatian IM Isolates															
B95-8		1					1	10							
China1					15				1					15	
Mediterranean	2		9			1				1		1	2	11	
North Carolina				4							4				1
aa Position	335	338	338	349	352	352	352	356	356	356	357	358	359	361	361
B95-8 Prototype ^1^	G	L	L	D	H	H	H	D	D	D	P	H	L	T	T
aa Change	S	S	P	A	S	R	N	H	A	N	S	P	V	M	R
Number of Cases in Croatian IM Isolates															
B95-8					3			3	1		2	2	1	2	
China1		13								1					1
Mediterranean	1	11		4	1	8									
North Carolina			4				4					4			

^1^ B95-8 prototype strain (Accession number: VO1555) was used as a reference amino acid sequence. ^2^ Amino acid position of aa change unique for Croatian IM isolates; aa = amino acid.

**Table 4 pathogens-12-00915-t004:** Selected demographic, clinical and virological parameters in patients with infectious mononucleosis.

Selected Parameters	Infectious Mononucleosis Patients (Number of Patients, Percentage)
Number of patients	33
Gender distribution (number of males, percentage)	15 males (45.5%)
Age (mean, range)	11.9 years (0.89–17.83)
Hospitalized patients (number, percentage)	7 (21.2%)
Days since symptom onset at clinical presentation (median, interquartile range)	5 (3–7)
Clinical symptoms (number, percentage)	
Pharyngitis	30 (90.1%)
Lymphadenitis	25 (75.8%)
Respiratory symptoms	21 (63.6%)
Hepatosplenomegaly	12 (36.4%)
Upper eyelid edema	6 (18.2%)
Periglandural edema	6 (18.2%)
Rash	3 (9.1%)
EBV LMP1 variant distribution	
Wild-type	14 (42.4%)
Wild-type/China1 co-infection	8 (24.2%)
North Carolina	4 (12.1%)
Wild-type/North Carolina co-infection	4 (12.1%)
Mediterranean	3 (9.1%)

EBV = Epstein–Barr virus, LMP1 = latent membrane protein 1.

## Data Availability

Data available on request from the corresponding author.

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
