# Peer review of "Distribution of Epstein–Barr Virus LMP1 Variants in Patients with Infectious Mononucleosis and Association with Selected Biochemical and Hematological Parameters"

_pathogens, 2023, doi:10.3390/pathogens12070915_

Round 1

Reviewer 1 Report

The study by Zidovec-Lepej et al presents data on the distribution of EBV LMP1 variants in a group of 73 Croatian children with IM. This study also sought to investigate a possible association between individual LMP-1 genetic variants found in their IM patients with selected clinical and laboratory parameters. The authors further examined these parameters by comparing patients with one variant versus co-variant infections.

Comments:

1.       Although this appears to be the largest study done on molecular analysis of EBV isolates in infectious mononucleosis patients, it should be acknowledged in the Discussion that the study is still small and that larger studies are needed to resolve whether there is an association between strain specificity and IM severity based on marker analysis.

2.       Was this a hypothesis-driven study? If so, then the hypothesis should be clearly stated.

3.       The authors aptly explain why they focused on LMP-1 gene variations. However, seeing that their analysis found no association with single variant infection, it might be appropriate to mention in the discussion that other viral factors may impact severity markers, such as EBV lytic gene expression and viral load.

4.       The blue dot graphs in Figure 2 seem to contain less than 51 dots representing patients infected with one variant. Can the authors confirm that all the patients are represented in these graphs?

Other comments:

-          Line 25: change “characteristical” to “characteristic”.

-          Line 31: change “Herpseviride” to “Herpesviridae

-          Line 70: change “as well as subsequent differentiation” to “and their subsequent transformation”.

-          Line 108 refers to EBV type 1, which I believe should be changed to EBV type 2.

-          Lines 127-128: should be deleted as they are repeated from lines 106-107.

-          Line 227: change “eight isolated” to “eight isolates”.

-          Line 368: change “we hypothesize” to “we speculate”.

-          Line 378: change “lower a” to “a lower”.

-          Lines 423-424: Please note that the second reference (Cao Y et al) spans reference numbers 2 and 3, which then causes all remaining references to become misaligned. Please correct.

Better proof-reading of the manuscript is required.

Author Response

Responses to comments by the Reviewer 1.

Dear reviewer 1, thank you for your helpful comments on how to improve our manuscript. Please find enclosed our corrections and responses that have been included in the revised version of the manuscript with track changes.

Comment 1.  Although this appears to be the largest study done on molecular analysis of EBV isolates in infectious mononucleosis patients, it should be acknowledged in the Discussion that the study is still small and that larger studies are needed to resolve whether there is an association between strain specificity and IM severity based on marker analysis.

Author’s response: In order to emphasise the issue raised by the reviewer, we included additional text in the discussion section describing the abovementioned limitations of our study as suggested.

Added text in section Discussion (lines 406-407): “However, further studies on larger IM cohorts are required to resolve the issue of a possible association between viral strain specificity and IM severity.

Comment 2.  Was this a hypothesis-driven study? If so, then the hypothesis should be clearly stated.

Author’s response: Determination of LMP1 gene variants in Croatian patients with IM represented an exploratory analysis. Consequently, we did not have specific expectations about the results of this analysis and did not hold any formal hypotheses regarding those results. Furthermore, considering that studies on possible association between EBV LMP1 variants and clinical manifestations of IM are scarce, we did not hold a specific hypothesis regarding those results either. The data provided in this study may prove useful in defining the hypotheses in future studies of EBV LMP1 variants.

Comment 3. The authors aptly explain why they focused on LMP-1 gene variations. However, seeing that their analysis found no association with single variant infection, it might be appropriate to mention in the discussion that other viral factors may impact severity markers, such as EBV lytic gene expression and viral load.

Author’s response: We included additional text in the Discussion section that explains the important point raised by the reviewer.

Revised/added text is underlined: lines 352-355,: We also evaluated the impact of EBV molecular diversity on selected clinical and laboratory parameters of IM patients and have shown, for the first time, important differences in immunological parameters in IM patients with respect to the coinfection with multiple EBV variants. Notably, we did not find a significant association between EBV LMP1 variants and IM clinical presentation. This could result from other viral factors impacting the clinical presentation of IM, such as lytic gene expression and viral load.

Comment 4. The blue dot graphs in Figure 2 seem to contain less than 51 dots representing patients infected with one variant. Can the authors confirm that all the patients are represented in these graphs?

Author’s response: Clinical and laboratory data were available for 33 IM patients, as stated in the manuscript (Section 2.1, Table 4). We confirm that Figure 2 represents all patients included in this part of the study.

Reviewer’s other comments and author’s responses:

-          Line 25: change “characteristical” to “characteristic” (corrected as suggested by the reviewer)

-          Line 31: change “Herpseviride” to “Herpesviridae” (corrected as suggested by the reviewer)

-          Line 70: change “as well as subsequent differentiation” to “and their subsequent transformation” (corrected as suggested by the reviewer)

-          Line 108 refers to EBV type 1, which I believe should be changed to EBV type 2 (a 150bp fragment corresponds to EBV type 2 so we corrected our mistake as suggested by the reviewer)

-          Lines 127-128: should be deleted as they are repeated from lines 106-107. (corrected as suggested by the reviewer)

-          Line 227: change “eight isolated” to “eight isolates”. (corrected as suggested by the reviewer)

-          Line 368: change “we hypothesize” to “we speculate”. (corrected as suggested by the reviewer)

-          Line 378: change “lower a” to “a lower”. (corrected as suggested by the reviewer)

-          Lines 423-424: Please note that the second reference (Cao Y et al) spans reference numbers 2 and 3, which then causes all remaining references to become misaligned. Please correct. (corrected as suggested by the reviewer)

Reviewer 2 Report

The study provides an analysis of 73 infectious mononucleosis patients in Croatia regarding what type of EBV they are infected with and whether there are any clinical parameters that could be associated with the infection of specific EBV types. The description of the dataset is well written. I have only a few minor suggestions for the improvement of the manuscript text.

Comments:

Line 72. Instead of “C-terminal part of LMP1 gene”, it should read “C-terminal part of LMP1 protein” or “3’ end of the LMP1 gene”.

While EBV was measured and sequenced in 73 IM patients, why the clinical information of only 33 patients of the 73 patients was used for analyzing the correlation between EBV variants and clinical features?

Line 107-108. “…497 bp specific for EBV type 1 and 150 bp 107 specific for EBV type 1…” One of them should be EBV type 2.

Line 109. “Amplification of the C terminus of the LMP1 gene…” It should read “amplification of the 3’end of the LMP1 gene”. The term C-terminus is used for proteins and not genes. I suggest checking this term throughout the manuscript.

It is well written. There are only a few typos and I suggested some text changes, which I have mentioned above.

Author Response

Responses to comments by the Reviewer 2

Dear reviewer 2, thank you for your helpful comments on how to improve our manuscript. Please find enclosed our corrections and responses that have been included in the revised version of the manuscript with track changes.

Reviewer's comment: Line 72. Instead of “C-terminal part of LMP1 gene”, it should read “C-terminal part of LMP1 protein” or “3’ end of the LMP1 gene”.

Author's response: Our mistake in the text was corrected according to the reviewer's instructions.

Reviewer's comment: Wile EBV was measured and sequenced in 73 IM patients, why the clinical information of only 33 patients of the 73 patients was used for analyzing the correlation between EBV variants and clinical features?

Author's response: Complete sets of selected clinical and laboratory data were extracted from the official hospital's database (UHID database) only for the selected 33 patients.Unfortunately, complete clinical and laboratory data were not available for the other 40 IM patients. As such, we did not include the stated patients in the part of the study dealing with the clinical presentation of IM.

Reviewer's comment:Line 107-108. “…497 bp specific for EBV type 1 and 150 bp 107 specific for EBV type 1…” One of them should be EBV type 2.

Author's response: We corrected the mistake in the text (the

 150 bp fragment corresponds to EBV type 2)

Reviewer's comment: Line 109. “Amplification of the C terminus of the LMP1 gene…” It should read “amplification of the 3’end of the LMP1 gene”. The term C-terminus is used for proteins and not genes. I suggest checking this term throughout the manuscript.

Author's mistake: We corrected the mistake in the text and checked for similar ones throughout the manucript.